# Potential Health Risks of Chemicals in Car Colorant Products

**DOI:** 10.3390/ijerph16060913

**Published:** 2019-03-14

**Authors:** Daeyeop Lee, Joo-hyon Kim, Moonyoung Hwang, Hyunwoo Lim, Kwangseol Seok

**Affiliations:** National Institute of Environmental Research, Hwangyeong-ro 42, Seo-gu, Incheon 22689, Korea; daeyub22@snu.ac.kr (D.L.); jhkim0318@korea.kr (J.-h.K.); myrang@korea.kr (M.H.); yaho365@korea.kr (H.L.)

**Keywords:** health risk, car colorant products, exposure assessment, risk assessment

## Abstract

Public concern regarding the use of products with chemicals has increased in Korea, following reports indicating that hazardous chemicals in products, such as disinfectants, can cause fatal lung disease. Despite the widespread use of car colorant products, little is known regarding their potential health risks. The purpose of this study was to determine the potential health risks of substances that exist in car colorant products. Thirteen car colorant products were purchased from the Korean market and 15 commonly used chemicals were analyzed. Exposure and risk assessments were conducted in two assessment stages (screening and refined). The analysis showed that all of the examined products contained toluene, ethylbenzene, and xylene. The maximum concentration of toluene was 52.5%, with a median concentration of 10.8%. Tier 1 (screening) assessment showed that four chemicals (toluene, ethylbenzene, xylene, and 2-butoxyethanol) may pose health risks, but tier 2 (refined) assessment showed that these chemicals do not pose any risk. However, these chemicals were present in all of the examined products, and government regulations did not control their concentrations in these products. Therefore, we suggest that levels of toluene, ethylbenzene, and xylene in car colorant products should be regulated to protect public health.

## 1. Introduction

Consumer products contain thousands of chemicals, some of which have been associated with developmental, reproductive, and other adverse health effects [1,2,3,4,5]. In 2011, an unidentified case of fatal lung disease was reported in Korea, which was probably caused by hazardous chemicals that are contained in disinfectant products [6]. As of 13 January 2017, hundreds of cases, resulting from this type of exposure, have been reported [7]. As a preventive measure, the Act on the registration and evaluation of chemicals in Korea (K-REACH) was passed on 30 April 2013 and then enforced on 1 January 2015 for the control and screening of hazardous chemicals, in order to protect the environment and public health. In addition, the Ministry of Environment in Korea (KME) (Abbreviations: ATSDR: Agency for Toxic Substances and Disease Registry, ECHA: European Chemical Agency, GC: gas chromatography, HQ: hazard quotient, KME: Ministry of Environment in Korea, KNIER: National Institute of Environmental Research in Korea, LOAEC: lowest observed adverse effect concentration, MOE: margin of exposure, NOAEC: no observed adverse effect concentration; NOEC: no observed effect concentration, RED: US Reregistration Eligibility Decision, MS: mass spectrometry, RfC: reference concentration, US EPA IRIS: United States Environmental Protection Agency Integrated Risk Information System) enforced Notice No. 2015-41 in April 2015, setting the regulations for the safety and labeling standards of eight product categories that were deemed to be of risk to human health and the environment [8]. As of 26 February 2018, 23 consumer products were classed as being risk-concerned products [9].

To determine the health risks, understanding product exposure is very important. Therefore, previous studies have been conducted to identify product usage patterns (e.g., frequency of use, amount of use per application, etc.) of national consumers, which are used as input parameters in exposure assessments [10,11,12,13]. The European Center for Ecotoxicology and Toxicology of Chemicals [14] and the Netherlands National Institute for Public Health [15] developed an exposure assessment tool for consumer products. In addition, the National Institute of Environmental Research in Korea (KNIER) also developed an exposure model for consumer products [16]. Recently, some studies have determined the exposure and health risks of substances in a number of consumer products. Liu et al. [17] determined the health risk of toxic metals in lip products in the United States and then suggested that some metals should be regulated. In addition, Guo and Kannan [18] reported human exposure to phthalates and parabens in personal care products. In Europe, the EPHECT project determined the exposure and risks of formaldehyde, d-limonene, α-pinene, and naphthalene in 15 consumer products [19,20]. In Korea, Kim et al. [21] determined the exposure levels and health risks of substances in deodorants. Furthermore, a study by Lim et al. [22] suggested that substances in consumer products exceed the safety limits and hence need to be reduced.

To protect public health from hazardous substances in consumer products, the lists of chemical safety limits and prohibited substances have to be regularly updated based on human health and environmental risks. In Europe, substances in biocidal products, including consumer products, were classed as approved, expired, under review, or not approved substances by the European Chemical Agency [23].

Recently, many car colorant products have been developed for consumer use, and they can easily be purchased in online and offline markets. Although car colorant products in Korea have been classed as being risk-concerned products, to the best of our knowledge, the potential health risks that are related to substances in these products have not been investigated. Moreover, only nine substances are currently managed, two (benzene and trichloroethene) have safety limits, and seven are prohibited by the Korean National Law Information Center [9]. Therefore, the health risks of substances in car colorant products need to be determined with some urgency. Therefore, the purpose of this study was to determine the potential health risks of substances in car colorant products.

## 2. Methods

### 2.1. Sample Collection and Target Chemical Analysis

The study samples were selected based on sales ranks at a big Korean shopping mall. A total of 13 products, which comprised four pen-based products, an emulsion product, and eight spray products, were purchased in 2017. The products were assumed to be commonly consumed by Korean people for general use only and not for occupational use, because the composition of chemicals in occupational use products might be different from that of the chemicals that are present in consumer products; the exposure factors are also significantly different.

To select the target substances for analysis, information on substances that are used in car colorant products was obtained from their manufacturing companies. Fifteen chemicals were selected for the study based on their frequency of use and their market share. The analytical method used for target chemicals is presented in Table 1. For all analyses, the standard operating procedure that was developed by KNIER was followed, and the quality assurance/quality control requirements were adhered to, including the method blank, reagent blank, instrument detection limit, and calibration curve [24].

### 2.2. Toxicological Information

The toxicological information of target chemicals was investigated using several steps. First, official toxicological reports and studies (e.g., United States Environmental Protection Agency Integrated Risk Information System (US EPA IRIS), US Reregistration Eligibility Decision (RED) report, EU ECHA registration dossier, etc.) for the target chemicals were reviewed. Second, the toxic value (e.g., lowest observed adverse effect concentration (LOAEC) and no observed adverse effect concentration (NOAEC)), reliability, experimental method (e.g., Organisation for Economic Co-operation and Development Test Guideline (OECD TG)), and the endpoints were identified in the toxicological data. If there were several toxicological values for a single chemical, the more conservative value was used. Moreover, the toxicological value from route-to-route exploration was excluded from the present study. Third, reference toxicological values were determined according to official ECHA guidance [25]. Uncertainties in the extrapolation of experimental data on the differences in exposure duration and differences in toxicological values were considered. For example, the uncertainty factor for exposure duration was adjusted to 2 (sub-chronic toxicological value to chronic value) or 6 (sub-acute toxicological value to chronic value). However, in cases where the toxicological US EPA IRS Reference Concentration (RfC) value is already adjusted to the chronic toxicological value for humans, the original RfC value was used as the reference toxicological value.

### 2.3. Exposure and Risk Assessment

To assess potential health risks, exposure was estimated using information obtained from KNIER [24]. Among the various types of exposure models, only the spray type was used to determine exposure (through inhalation) in this study, because models for pen and emulsion exposure types could not be found. Dermal and oral exposures were not determined, since oral exposure does not occur during the normal use of car colorant products, and the toxicity reference value for dermal exposure is very limited. Exposure was measured in two assessment stages: screening and refined (Figure 1). Moreover, exposure was based on single product use. The tier 1 assessment is widely used to screen consumer exposure and it is based on the summation of the highest percentile of product consumption, amount per use, and concentration of substances, while assuming the worst-case exposure scenario [26]. The tier 2 assessment is used to estimate refined consumer exposure levels [21].

The margin of exposure (MOE) and the hazard quotient (HQ) were used to determine the health risk. MOE is defined as the ratio of NOAEC for critical effect to theoretical, predicted, or estimated dose or concentration [27]. HQ is defined as the ratio of exposure dose to reference dose. The target MOE was established using the official ECHA guidance, and the multiplication of inter- and intra-species factors was considered [25]. For example, the inter-species extrapolation for rat was 2.5 and an intra-species variation of 10 was used to determine the target MOE. If the target chemical already had an RfC value that was adjusted to the toxicological value for humans, the target MOE was not calculated, and the HQ was used to determine the health risk. If the MOE for the target chemical is lower than the target MOE, and if the HQ is higher than 1, then the chemical may pose a health risk. The purpose of the tier 1 assessment was to screen for health risk, and a screening factor of 10 was used to achieve this. In the tier 1 assessment, the target MOEs were 10 times higher than normal MOEs, and the target HQs were 10 times lower than normal HQs. If the target chemical posed a health risk at this stage, the chemical was subjected to tier 2 assessment. If a chemical did not have toxicological information or it was not detected, tier 2 assessment was not conducted.

#### 2.3.1. Tier 1 Assessment

On the basis of the maximum concentrations of chemicals in the examined products and the 95th percentile consumer exposure factors (Table 2), the exposure concentration was calculated, as shown in Equation (1):*C*_a_ = *A*_p_·*W*_f_/*V**C*_inh_ = *C*_a_·abs·*t*·*n*/24(1)
where *C*_a_ is the target chemical concentration in the air (mg/m^3^), *A*_p_ is the amount of product used (mg), *W*_f_ is the fraction of the substance in the product, *C*_inh_ is exposure concentration via inhalation (mg/m^3^), *t* is the duration of product use (h), *n* is the frequency of product use (event/day), and *V* is the volume of space (m^3^). Assuming the worst-case scenario, when considering that most products are used indoors, *W*_f_ will be the maximum concentration of chemicals. Abs (absorption ratio to body) will be assumed to be 100%. *V* will be assumed to be 9.3 m^3^, which is the mean size of a bathroom in Korea, as reported in the Korean Consumer Exposure Factors [16].

#### 2.3.2. Tier 2 Assessment

On the basis of the concentrations of chemicals in the products in this study and the 75th percentile consumer exposure factors (Table 2), exposure concentration was calculated, as shown in Equation (2):*C*_a_ = (*A*_p_·*W*_f_/*V*·*N*)·[1 − exp(−*N*·*t*)]/*t*·AF*C*_inh_ = *C*_a_·abs·*t*·*n*/24(2)
where *N* is ventilation rate (h^−1^) and it is assumed to be 0.5 h^−1^, which is the general ventilation rate in Korean indoor environments [16]. Abs, *V*, and *W*_f_ are the same values as in the tier 1 assessment. AF is airborne faction and it is assumed to be 1 [28].

## 3. Results

### 3.1. The Concentration of Target Chemicals in Car Colorant Products

The detection rate and the range of chemical concentrations are shown in Table 3. Toluene, ethylbenzene, and xylene were detected in all examined products, with toluene having the highest concentration out of all the examined chemicals. Among the target chemicals, 10 chemicals were detected in over 50% of products. The concentrations of target chemicals substantially varied across products. For example, the highest concentrations of toluene and butyl acetate were 52.524% and 27.457%, respectively, but their lowest concentrations were only 0.055% and 0.017, respectively.

### 3.2. Toxicological Information

An evaluation of the toxicological values was carried out for reliability, exposure time, exposure duration, exposure route, and other factors. The summary of toxicological information for each target chemical is presented in Table 4. Among the 15 target chemicals, 13 had inhalation toxicological values. The US EPA IRIS system was used for six chemicals (methyl ethyl ketone, methyl isobutyl ketone, toluene, ethylbenzene, xylene, and 2-butoxyethanol). The ECHA registration dossier was used for another six chemicals (hexane, ethyl acetate, heptane, butyl acetate/*N*-butyl acetate, and propylene glycol methyl ether acetate). Toxic Substances and Diseases Registry (ATSDR) toxicological profile obtained the toxicological information of acetone. However, inhalation toxicological information for cyclohexanone and 2-butoxyethanol acetate was not found. The toxicological values from ECHA and ATSDR were adjusted to reference values (chronic reference toxicological values for humans). The calculated target MOE and HQ of the chemicals in car colorant products are shown in Table 5.

### 3.3. Exposure Assessment and Risk Characterization

#### 3.3.1. Tier 1 Assessment

A summary of the tier 1 exposure assessment is presented in Table 6. According to the assessment results, four chemicals (toluene, ethylbenzene, xylene, and 2-butoxyethanol) were identified as posing a health risk. The highest exposure concentration was that of toluene (0.89 mg/m^3^), because it is the chemical that contains the highest concentration in the examined products (Table 3). The exposure concentration of xylene was only 0.26 mg/m^3^ (which was not the highest), but its HQ was the highest at 2.6, because xylene had the lowest reference toxicological value out of all the chemicals (Table 4).

#### 3.3.2. Tier 2 Assessment

A summary of the tier 2 exposure assessment is shown in Table 7. The HQs of all the selected target chemicals were lower than the target HQs (Table 5); therefore, the substances may not pose a health risk. The HQ for 2-butoxyethanol was much lower than the target HQ. However, the HQ for xylene was only 62% lower than the target HQ, because of the very low reference toxicological value (Table 4).

## 4. Discussion

This preliminary study of chemicals that are present in car colorant products suggested that they might pose potential health risks. Studies both in Korea and in other countries have reported health risks for chemicals in the workplace and in indoor and outdoor environments [29,30,31,32,33]. Our understanding of how chemical concentrations in consumer products may translate to potential health risk is still changing. Recently, a few studies were conducted to determine the chemicals that were used in consumer products. Lim et al. [22] investigated 207 consumer products (e.g., shoe polishes, oil-based ballpoint pens, pencils, glues, and other products) in local Korean markets. These were analyzed for benzene, toluene, ethylbenzene, and xylene, which were found to be present in 28.5% of products. In the present study, all of the examined products contained toluene, ethylbenzene, and xylene. Moreover, the concentrations of all three chemicals were higher than those of other chemicals and the maximum concentration of toluene was around 52.5% (Table 2). Thus, these three chemicals might be widely used as solvents in car colorant products.

To determine health risks related to using consumer products, toxicological information is necessary. Several scientific studies have shown that inhalation exposure to chemicals is associated with adverse health effects [34,35,36]. For example, one study reported that both acute and chronic toluene inhalation affected the nervous system [37]. In addition, the International Agency for Research and Cancer [38] categorized ethylbenzene as a Group 2B compound (possibly carcinogenic in humans). In addition, other chemicals in car colorant products might also be associated with adverse health effects. However, the toxicological information for the chemicals in consumer products is still incomplete. For example, 23.1% of examined products in this study contained cyclohexane and 2-butoxyethanol, but official inhalation toxicological information (e.g., US EPA IRIS, RED report, ECHA registration dossier, etc.) for these chemicals was not found. Thus, the health risk for these two chemicals could not be determined, and further studies on their inhalation toxicological values are required with some urgency.

Exposure assessment is a crucial tool in the prevention of public exposure to hazardous chemicals in products. Deterministic methods use point estimates of input parameters to provide a single worst-case value [39]. Previous studies have used deterministic methods to screen consumer products exposure levels [18,20,21,40]. To use this method to assess exposure to chemicals in consumer products, information on exposure factors, such as frequency of use, amount of use per application, as well as information about the circumstance of use, are required [41]. The usage patterns of consumer products may differ between countries, so data on national exposure factors are necessary. In Korea, the KME developed a national exposure database (the National Consumer Exposure Factors), which was made available to the public. The values were set up as the 5th, 50th, 75th, and 95th percentiles [8,9]. Therefore, the results of the exposure assessment that was conducted in the present study showed a reasonable Korean consumer exposure level. A recent study on consumer products used two assessment stages to determine the health risks: screening and refined assessments [21]. The screening assessment (tier 1) is used to prioritize chemicals before any further risk assessment. The present study also used two stages for assessment (tier 1 and tier 2). After the tier 1 assessment, four chemicals (toluene, ethylbenzene, xylene, and 2-butoxyethanol) were found to pose adverse health risk. However, the tier 2 assessment showed that none of the chemicals may be of risk. This is because the tier 2 assessment considered the ventilation rate and the general consumer usage patterns. The exposure concentrations in the tier 2 assessment were compared with international indoor air quality guide values published in Germany (Table 8 [42]). Guide values II are effect-related values that are based on current toxicological and epidemiological information, and concentrations over these values may pose a health hazard. Guide values I are chemical concentrations that, at present, have not been found to cause adverse health effects, even with life-long exposure [42]. The exposure concentrations in the tier 2 assessment of toluene, ethylbenzene, xylene, and 2-butoxyethanol were much lower than the indoor guide values. However, these chemicals were not regulated in car colorant products, meaning that they can be used in consumer products in the Korean market without any restrictions. When the chemical safety limit regulation list is updated by the KME, these three chemicals should be prioritized for review.

As multiple consumer products are used in everyday life, the consumers can be exposed to specific chemicals by accumulation through using different products. To assess the summation of chemical exposure, the co-use patterns of consumer products have been investigated. Garcia-Hidalgo et al. [12] reported that females and males in Switzerland used 5.47 ± 2.21 and 5.88 ± 2.73 cleaning products, respectively, and also assessed product combinations. Another study investigated the co-use patterns of cosmetic and skin care products, and the summed aggregate exposure was calculated for the everyday use of multiple consumer products [43]. Shin et al. [44] established a screening model framework that integrates the exposure to chemicals in cleaning products from all plausible exposure pathways that result from indoor residential use. This means that, although the exposure concentration of a single product use may not pose any health risks (as shown in the tier 2 assessment in this study), the accumulated exposure from the use of multiple consumer products can have adverse health effects for the consumer. Therefore, future studies should consider analyzing the co-use patterns for various consumer products to determine the total substance exposure.

The present study had several strong points. First, the actual concentrations in car colorant products and national consumer exposure factors were used to assess health risks. Second, the examined 15 chemicals were selected while using manufacturer information; hence, the health risks of widely used chemicals in car colorant could be determined. This study also had some limitations. When the exposure assessment was conducted, the product application type was not considered. To the best of our knowledge, there are no exposure models for pen and emulsion product types. However, as the exposure model for spray products was not underestimated, the results from this study can be considered to be acceptable. Another limitation was the lack of information on the co-use of car colorant products, as this study was focused on single use only. In conclusion, future studies should be conducted to determine the health risks of multiple car colorant product use.

## 5. Conclusions

To the best of our knowledge, this is the first study to determine the health risk of chemicals in car colorant products. We found that the 15 examined chemicals may not pose health risks. However, toluene, ethylbenzene, and xylene were detected in all of the examined products at high concentrations, and they do not have any imposed restrictions. To protect public health, regulatory institutions should not only assess the presence of hazardous substances, but also compare estimated exposure with toxicological values. Therefore, the results of this study suggest that toluene, ethylbenzene, and xylene should be regulated in car colorant products.

## Figures and Tables

**Figure 1 ijerph-16-00913-f001:**
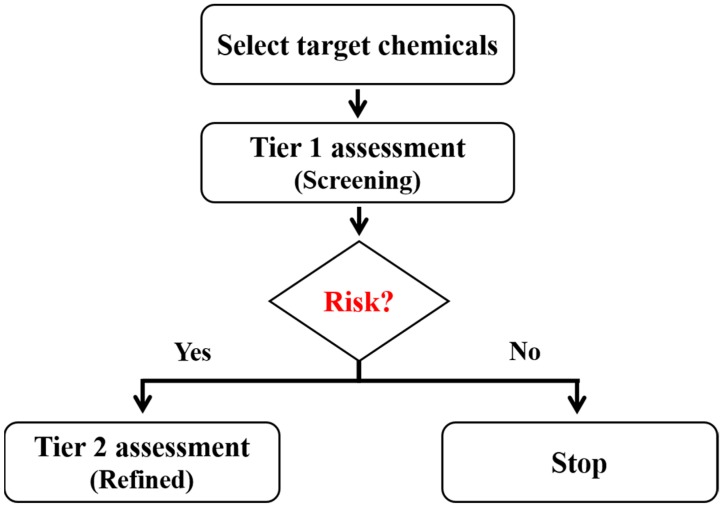
Workflow for tier 1 and 2 assessments.

**Table 1 ijerph-16-00913-t001:** The analytical method used for target chemicals.

Chemicals	Pre-Treatment Method	Analytical Method *	Limit of Quantitation (% *w*/*w*)
Methyl ethyl ketone, Ethyl acetate, Isobutanol, Heptane, Methyl isobutyl ketone, Toluene, Butyl acetate/*N*-butyl acetate, Ethylbenzene, Xylene, Propylene glycol methyl ether acetate, 2-Butoxyethanol, Cyclohexanone, 2-Butoxyethanol acetate	Solvent extraction	GC/MS	0.001
Acetone, Hexane	Solvent extraction	GC/MS	0.005

* Gas chromatogram phy (GC), mass spectrometry (MS).

**Table 2 ijerph-16-00913-t002:** Input parameters for exposure assessment.

Exposure Factors	Percentiles	
	75th	95th
Frequency of use (events/6 months)	2.00 ^a^	3.00 ^a^
Duration of use (min/use)	20.00 ^a^	50.00 ^a^
Amount of use per application (g/use)	13.60 ^a^	27.21 ^a^

^a^ Korean consumer exposure factors for car colorant products from the Korean National Law Information Center [16].

**Table 3 ijerph-16-00913-t003:** Detection rate and concentration (median, minimum, and maximum) of target chemicals in car colorant products.

Chemical	Detection Rate (%)	Concentration Range (%)
Acetone	61.5	0.011 (0.006, 3.088)
Hexane	38.5	0.486 (0.024, 0.932)
Methyl ethyl ketone	61.5	1.373 (0.031, 4.102)
Ethyl acetate	38.5	0.833 (0.002, 4.991)
Isobutanol	46.2	0.129 (0.022, 1.207)
Heptane	53.8	0.161 (0.003, 2.466)
Methyl isobutyl ketone	61.5	0.053 (0.002, 2.452)
Toluene	100.0	10.804 (0.055, 52.524)
Butyl acetate; *N*-butyl acetate	69.2	17.676 (0.017, 27.457)
Ethylbenzene	100.0	2.722 (1.090, 24.381)
Xylene	100.0	4.880 (0.853, 15.492)
Propylene glycol methyl ether acetate	62.9	1.868 (0.205, 11.410)
2-Butoxyethanol	38.5	2.352 (1.391, 3.597)
Cyclohexanone	23.1	0.015 (0.002, 0.726)
2-Butoxyethanol acetate	23.1	0.012 (0.004, 0.983)

**Table 4 ijerph-16-00913-t004:** Summary of toxicological information for target chemicals.

Chemical	Toxicity Value (NOAEC, NOEC, LOAEC; mg/m^3^)	End Point	Adjustment Factors (Exposure Time)	Assessment Factors (Exposure Duration)	Reference Toxicological Value (mg/m^3^)
Acetone ^c^	LOAEC = 2952	Neurological effects	-	LOAEC to NOAEC: 3	984.7
Hexane ^b^	LOAEC = 10,574	Nasal lesions	0.50	Sub-chronic to chronic: 2 LOAEC to NOAEC: 3	881.2
Methyl ethyl ketone ^a^	-	Developmental toxicity (skeletal variations)	-	-	5.0
Ethyl acetate ^b^	NOAEC = 1280	Respiratory irritant effects	0.18	Sub-chronic to chronic: 2	114.3
Isobutanol ^b^	NOAEC = 7500	No effects observed	0.18	Sub-chronic to chronic: 2	669.6
Heptane ^b^	NOAEC = 12,470	Body weight changes	0.50	Sub-chronic to chronic: 2	3117.5
Methyl isobutyl ketone ^a^	-	Reduced fetal body weight, skeletal variations, and increased fetal death	-	-	3.0
Toluene ^a^	-	Neurological effects	-	-	5.0
Butyl acetate; *N*-butyl acetate ^b^	NOAEC = 2400	Organ-specific toxicity	0.18	Sub-chronic to chronic:2	214.3
Ethylbenzene ^a^	-	Developmental toxicity	-	-	1.0
Xylene ^a^	-	Impaired motor coordination	-	-	0.1
Propylene glycol methyl ether acetate ^b^	NOEC = 1621	Increase in tumor incidence	0.18	Chronic to chronic: 1	289.6
2-Butoxyethanol ^a^	-	Hemosiderin deposition in the liver	-	-	1.6
Cyclohexanone	N/A	N/A	N/A	N/A	N/A
2-Butoxyethanol acetate	N/A	N/A	N/A	N/A	N/A

^a^ United States Environmental Protection Agency, Integrated Risk Information System (US EPA IRIS). ^b^ European Chemicals Agency (ECHA), registration dossier. ^c^ Agency for Toxic Substances and Disease Registry (ATSDR), toxicological profile. No observed adverse effect concentration (NOAEC), no observed effect concentration (NOEC), lowest observed adverse effect concentration (LOAEC).

**Table 5 ijerph-16-00913-t005:** Summary of Margin of exposure (MOE) and hazard quotient (HQ) for target chemicals from tier 1 and 2 assessments.

Chemical	Target MOE/HQ	
	Tier 1 Assessment	Tier 2 Assessment
Acetone	100 Intra-species: 10 Screening factor 10	10 Intra-species: 10
Hexane	250 Intra-species: 10 Inter-species: 2.5, Screening factor 10	250 Intra-species: 10 Inter-species: 2.5
Methyl ethyl ketone	0.1 *	1 *
Ethyl acetate	250 Intra-species: 10 Inter-species: 2.5, Screening factor 10	25 Intra-species: 10 Inter-species: 2.5
Isobutanol	250 Intra-species: 10 Inter-species: 2.5, Screening factor 10	25 Intra-species: 10 Inter-species: 2.5
Heptane	250 Intra-species: 10 Inter-species: 2.5, Screening factor 10	25 Intra-species: 10 Inter-species: 2.5
Methyl isobutyl ketone	0.1 *	1 *
Toluene	0.1 *	1 *
Butyl acetate; *N*-butyl acetate	250 Intra-species: 10 Inter-species: 2.5, Screening factor 10	25 Intra-species: 10 Inter-species: 2.5
Ethylbenzene	0.1 *	1 *
Xylene	0.1 *	1 *
Propylene glycol methyl ether acetate	250 Intra-species: 10 Inter-species: 2.5, Screening factor 10	25 Intra-species: 10 Inter-species: 2.5
2-Butoxyethanol	0.1 *	1 *
Cyclohexanone	N/A	N/A
2-Butoxyethanol acetate	N/A	N/A

* Hazard quotient (HQ).

**Table 6 ijerph-16-00913-t006:** Tier 1 assessment results.

Chemical	Exposure Concentration (mg/m^3^)	MOE or HQ ^a^	Health Risk (O = yes, X = no)
Acetone	0.05	18,830,375 *	X
Hexane	0.02	55,864,263 *	X
Methyl ethyl ketone	0.07	0.01 **	X
Ethyl acetate	0.08	1,352,490 *	X
Isobutanol	0.02	32,759,583 *	X
Heptane	0.04	74,658,040 *	X
Methyl isobutyl ketone	0.07	0.01 **	X
Toluene	0.89	0.2 **	O
Butyl acetate; *N*-butyl acetate	0.46	460,935 *	X
Ethylbenzene	0.41	0.4 **	O
Xylene	0.26	2.6 **	O
Propylene glycol methyl ether acetate	0.19	1,498,850 *	X
2-Butoxyethanol	0.44	0.3 **	O
Cyclohexanone	N/A	N/A	N/A
2-Butoxyethanol acetate	N/A	N/A	N/A

^a,^* Margin of exposure (MOE), ** hazard quotient (HQ).

**Table 7 ijerph-16-00913-t007:** Tier 2 assessment results.

Chemical	Exposure Concentration (mg/m^3^)	HQ ^a^	Health Risk (O = yes, X = no)
Toluene	0.13	0.03	X
Ethylbenzene	0.06	0.06	X
Xylene	0.04	0.38	X
2-Butoxyethanol	0.01	0.01	X

^a^ Hazard quotient (HQ).

**Table 8 ijerph-16-00913-t008:** Comparison between exposure levels current study and international indoor guide values.

Chemical	International Indoor Guide Values *		Current Study
	II	I	
Toluene	3.0 mg/m^3^	0.3 mg/m^3^	0.13 mg/m^3^
Ethylbenzene	2.0 mg/m^3^	0.2 mg/m^3^	0.06 mg/m^3^
Xylene	0.8 mg/m^3^	0.1 mg/m^3^	0.04 mg/m^3^
2-Butoxyethanol	1.0 mg/m^3^	0.1 mg/m^3^	0.01 mg/m^3^

* Indoor guide values generated by a German committee [42].

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
