# Peer review of "Potential Health Risks of Chemicals in Car Colorant Products"

_ijerph, 2019, doi:10.3390/ijerph16060913_

Round 1
Reviewer 1 Report
With the described methods, it is difficult to tell exactly how the literature review was conducted and referenced in terms of developing risk assessments for tier 1. Please provide further more in depth description. For example, what references were used to develop criteria (line 87).
The description of equation 1 lists volume assumed to be mean size of a bathroom (line 137). Why use bathroom area for assessing car colorants? Please revise, or, if car colorants are used in a bathroom, please provide details of use.
Introduction and conclusions reference multiple specific examples of exposure scenarios for a myriad of chemicals (lines 52, 57, 249). How do these relate to car colorants?
Study would benefit from assessment additional exposure routes (e.g., ingestion of aerosol particulates) and application types.
Call for toxicological data on cyclohexane and 2-butoxyethanol acetate? These were left out of assessment due to lack of information.
Since the only differences between chemicals outlined in Table 1 is the detection limit, this table may be explained in text.
Please revise blank page after Table 4.
Please define use of single asterisk in Table 6.
Statements within results and conclusion sections are at odds with one another. For example, lines 193 and 271 present opposing findings.
Author Response
Thank you for your valuable comments.
I revised my paper by your advice.
And also english is rechecked by native english editor.
Thank you.

Reviewer 2 Report
In this manuscript, the authors analyzed that the concentrations of 15 chemicals in 13 car colorant products and investigated whether these chemicals potentially had health risks using databases. Since the car colorants were easily available and widely used in lots of countries as well as Korea, this study includes important information about the potential health risk of these chemicals.
This study was well designed, and the manuscript was well written. However, there are several points that should be modified.
Minor comments
#1. To reviewer’s understanding, the authors analyzed 13 products in the study. However, in lines 70-72, there are only 11 products are shown (2 of pen-based products, one emulsion product, and eight spray products). Could the authors explain the number of products examined in the study?
#2. In the third column of Table 6, the reviewer thinks “*” and “**” indicate MOE and HQ, respectively. However, the explanation of “a” is shown as “Margin of exposure (MOE), **hazard quotient (HQ)”. The reviewer thinks the authors should add “*” just before “Margin of exposure (MOE)”. The reviewer also thinks the title of the third column should be “MOE or HQ a”.
Author Response

(The authors gave the same response as above.)

Reviewer 3 Report
Interesting and important paper.
More specific information is needed regarding some of the background info - see comments. Also, since this is a new REACH related legislation, it would be interesting to indicate briefly its similarity/difference to EU REACH, for example was it adapted to the situation in Korea or is it fundamentally different in some regards.
Please see comments in text in general and especially pertaining to confusion in Table 5.
While you acknowledged no information is available regarding product application in this paper, you didn't explain why you used a volume typical of the size of a Korean bathroom, is this because products would be stored here? Would persons be expected to apply products indoors or outdoors?
Further information on the potential health impacts of toluene, ethylbenzene, and xylene should be included and if these chemicals are known to cause occupational health impacts in applying car colorants or during other use, as often industrial or workshop quantities of hazardous chemicals are sold in consumer products but in smaller quantities. Also, if workers are required to use protective equipment and conditions before applying these chemicals.
Author Response
Thank you for your valuable comments.
I revised my paper by your advice.
(english is also rechecked by native english editor.)
You mentioned study design is not appropriate.
Howevere, lots of study used this method (reference 21, 25, 26, 27 etc). So I believed this design is already verified.
Thank you.
